# Transferrin Mediated NCC Killing Activity through NCCRP-1 in Nile Tilapia (*Oreochromis niloticus*)

Yu Huang [1,2,3], Zhengsi Chen [1,2,3], Ruitao Xie [4,5], Pei Wang [6], Zhiqiang Zhang [1,2,3], Jia Cai [1,2,3], Bei Wang [1,2,3] and Jichang Jian [1,2,3,*]

1 Guangdong Provincial Key Laboratory of Aquatic Animal Disease Control and Healthy Culture, College of Fishery, Guangdong Ocean University, Zhanjiang 524000, China
2 Laboratory for Marine Biology and Biotechnology, Qingdao National Laboratory for Marine Science and Technology, Qingdao 266000, China
3 Guangdong Provincial Engineering Research Center for Aquatic Animal Health Assessment, Shenzhen 327005, China
4 Key Laboratory of Aquatic, Livestock, and Poultry Feed Science and Technology in South China, Ministry of Agriculture and Rural Affairs, Zhangjiang 524000, China
5 Guangdong Evergreen Feed Industry Co., Ltd., Zhangjiang 524000, China
6 Guangxi Key Laboratory of Beibu Gulf Marine Biodiversity Conservation, Beibu Gulf University, Qinzhou 535000, China
* Correspondence: jianjc@gdou.edu.cn

**Abstract:** Non-specific cytotoxic cell l (NCC) is a kind of important lymphocyte participating in the non-specific immune response in teleost. Non-specific cytotoxic cell receptor protein 1 (NCCRP-1) is a receptor molecule on the surface of NCC and plays an important role in mediating the activity of NCC. However, there are few reports on which signal molecule could transmit signals through NCCRP-1. In this study, yeast two-hybrid library of tilapia liver and head kidney was constructed, and a transferrin from *Oreochromis niloticus* (On-TF) with interaction protein sequence was obtained by screening the library with bait vector NCCRP-1 of *Oreochromis niloticus* (On-NCCRP-1). Then, the open reading frame (ORF) of On-TF was cloned, which had 2088 bp in length, encoding polypeptides of 695 amino acids. The deduced amino acid sequence was highly homologous to teleost and similar to mammalian TF, containing two TR_FER domains (25-343th aa and 344-686th aa) for binding iron ions. Furthermore, a point-to-point yeast two-hybrid method was used to further verify whether On-NCCRP-1 could bind to On-TF. The recombinant On-TF (rOn-TF) protein was purified by prokaryotic expression system. In vitro experiments showed that rOn-TF could up-regulate the expression of killing effector molecule of NCC by On-NCCRP-1, and rOn-TF-activated NCCs showed a significantly improved ability to kill FHM cells, indicating that rOn-TF could regulate the NCC signaling pathway through NCC receptor molecule On-NCCRP-1. The results provide a more theoretical basis for understanding the regulation mechanism of NCC activity.

**Keywords:** aquaculture; cytotoxicity; immune response; natural killer cell; apoptosis

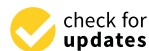



## 1. Introduction

Mammalian natural killer (NK) cells are important effector cells of the innate immune system, as a result of its non-specific cytotoxic killing effect [1]. Similar to mammals' NK cells, cytotoxic cells are also found in fish, which are called non-specific cytotoxic cells (NCCs). NCCs are the first line of defense in the process of fish anti-viral, anti-bacterial, anti-parasitic, and anti-tumor responses, and are thus known to play critical role in fish innate immunity [2]. NCCs are therefore considered as the precursor cell of NK cells evolutionarily present in fish [3].

In 1984, the function of NCCs—observed as killing transplanted human or mouse cell line cells—was found in channel catfish (*Ictalurus punctatus*) [4–7]. Since then, NCCs

have also been found in a variety of fish, such as tilapia (*Oreochromis aureus*), rainbow trout (*Oncorhynchus mykiss*), carp (*Cyprinus carpio*), river red salmon (*Salvelinus continulus*), and orange-spotted grouper (*Epinephelus coioides*) [7–10]. NCCs can specifically recognize different antigen ligands through non-specific cytotoxic cell receptor protein-1 (NCCRP-1) on the surface to activate its cytotoxic effect [11]. Activated NCCs can induce apoptosis of target cells, similar to the killing mechanism of mammalian NK cells. The killing effect of NCCs is also achieved through granzyme and Fas or Fas ligands (FasL) [12]. Activated NCCs can release cytotoxic-related factors such as granzyme, perforin, and granlysin to kill target cells through immune synapses. At the same time, NCCs can also produce a large number of FasL, which can be released upon activation and bind to Fas receptors on the surface of target cells to induce apoptosis [13].

Many types of receptors have been identified on the surface of NK cells. These receptors regulate the function of NK cells by interacting with different ligands [14]. NCCRP-1 is an important type III transmembrane protein on NCC and belongs to lectin-type receptor [15]. Meanwhile, NCCRP-1 is considered as a specific surface marker of NCCs. NCCRP-1 cross-linking can phosphorylate receptor tyrosine and serine, thereby activating the downstream innate immune response. However, there is no relevant report on whether NCCRP-1 can bind with other interacting proteins in vivo and then transmit signals to regulate the killing activity of NCCs. Therefore, Nile tilapia, an important fish for breeding, was selected as the research object in this experiment. Firstly, the interaction protein of NCCRP-1 in tilapia (On-NCCRP-1) was obtained by screening yeast two-hybrid (Y2H) library. Then, the interaction protein gene was cloned and characterized, and the regulation of NCC activity after interaction with NCCRP-1 was further analyzed. The results provide more data support for understanding the role of NCCs in non-specific immune responses in fish.

## 2. Materials and Methods

### 2.1. Fish Rearing and Sample Collection

All Nile tilapia (50 ± 5 g) were procured from a commercial fish farm in Zhanjiang city, Guangdong Province, China. The same methods as in earlier studies were used in rearing the fish [16]. Briefly, the fish were maintained at room temperature in aerated freshwater and acclimatized in 1000 L tanks at a stocking rate of 4 g/L at 28 ± 2 °C for 2 weeks, during which they were fed with commercial feed daily (3% body weight). The dissolved oxygen and pH were carefully maintained in the ranges of 5.0–6.0 mg/L and 7.3–7.8, respectively. At the cessation of the experiment, the healthy fish were anesthetized with MS-222 (Sigma, Darmstadt, Germany), after which their organs (liver and head kidney) were collected. The collected samples were immediately frozen in liquid nitrogen and stored at −80 °C until use. All experiments were conducted following the Guangdong Province laboratory animal management regulations and approved by the Ethics Committee of the Guangdong Ocean University (Date: 10 May 2019).

### 2.2. RNA Extraction and cDNA Synthesis

RNA from liver and head kidney above extraction and cDNA synthesis were done in accordance with our previous study [17]. Briefly, Total RNA were extracted using TransZol Up (Transgen, Beijing, China) according to the manufacturer's protocol. The quality of total RNA was detected by electrophoresis on 1% agarose gel. EasyScript® One-Step gDNA Removal and cDNA Synthesis SuperMix (TaKaRa, Dalian, China) were selected to synthesize cDNA.

### 2.3. Construction of pGBKT7-On-NCCRP-1 Plasmid

In order to construct the bait plasmid of pGBKT7-On-NCCRP-1 for Y2H, a primer pair was designed to amplify the complete On-NCCRP-1 gene according to the sequence published by NCBI (GenBank: MF162296.1). Restriction sites for Nde *I* and Bam H*I* (underlined) were created at the 5′-ends of the upstream (On-NCCRP-1-S) and downstream

(On-NCCRP-1-A) primers, respectively, to facilitate subsequent directional cloning of the On-NCCRP-1 gene into the pGBKT7 vector (Clontech, San Francisco, CA, USA). The correct orientation of the insert in the pGBKT7-On-NCCRP-1 plasmid was verified through sequencing. All polymerase chain reaction (PCR) primers used in this study are listed in Table S1. The gene accession number is shown in Supplementary Table S2.

### 2.4. Construction of the Liver and Head Kidney cDNA Y2H Library

In order to construct the Nile tilapia liver and head kidney cDNA library for Y2H, the total RNAs of Nile tilapia were extracted from liver and head kidney tissue using RNAiso reagents, and mRNA was obtained using Oligotex-dT30<super> (TaKaRa, Dalian, China). A cDNA library was then constructed as described in the Matchmaker™ Library Construction & Screening Kits User Manual (Clontech), and double-stranded cDNA and pGADT7-Rec vector were co-transformed into Saccharomyces cerevisiae Y187.

### 2.5. Screening the cDNA Library by Yeast Mating

pGBKT7-On-NCCRP-1 was transformed into *S. cerevisiae* Y187, and the cDNA library for Y2H was screened by yeast mating according to the manufacturer's protocols. Plasmids were prepared from positive clones and used to transform *Escherichia coli* strain JM109. Plasmids were recovered from yeast using the Yeastmaker Yeast Plasmid Isolation kit (Clontech). Then, they were transformed into *E. coli* strain DH 5$\alpha$ on Luria-Bertani (LB) agar plates containing ampicillin. The recovered plasmids were purified and sequenced. Diploids were plated on triple dropout medium (low stringency selection), and positives were re-plated onto quadruple dropout plates (high stringency) containing X-a-gal. The target gene was subsequently reinserted into the pGADT7 vector, and the Y2H assay was repeated. All the positive interactions were re-tested using yeast mating to eliminate false positives.

### 2.6. Cloning and Bioinformatics Analysis of On-TF

Total RNA from the head kidney was extracted using the EasyPure RNA Kit (Trans-Gen, Beijing, China). First-strand cDNA was synthesized from total RNA. First-strand cDNA served as the template for amplifying the partial cDNA sequence of On-TF through PCR using specific primers designed in NCBI data. The full-length cDNA sequence was synthesized using PCR technology as in a previous study [18].

### 2.7. Verification of On-TF and On-NCCRP-1 Interaction

In this study, the potential interaction proteins for On-TF and On-NCCRP-1 were screened from the library in the early stage (NCCRP-1 was linked to the PGBKT7 vector, and the library sequence was linked to pGADT7). In order to further verify the interaction between On-NCCRP-1 and On-TF, the Y2H technology was further used for point-to-point verification of the screened proteins, On-NCCRP-1 was used to connect with the pGADT7 vector, and On-TF was used to connect with the PGBKT7 vector. Meanwhile, a corresponding control group was set up to exclude false positives.

In brief, the sequences of On-TF and On-NCCRP-1 genes were amplified and sub-cloned into the plasmids of pGBKT7 and pGADT7 (Clontech), respectively. The experimental group consisted of AH109 yeast strain co-transfected with PGBKT7-On-TF and pGADT7-On-NCCRP-1 plasmids. The AH109 co-transfected with pGBKT7-53 and pGADT7-T plasmids served as a positive control, and that co-transfected with pGBKT7-Lam and pGADT7-T plasmids served negative control. For self-activation detection, AH109 was separately co-transfected with pGADT7-On-NCCRP-1 and pGBKT7 plasmids, as well as pGBKT7-On-TF and pGADT7 plasmids. Yeasts were grown on plates with a double dropout medium (SD/-Leu/-Trp) for 3–5 days at 30 °C. The positive monoclonal colony was selected for the five groups above and then separately spread on the plates containing quadruple dropout medium (SD/-Leu/-Trp/-His/-Ade) and quadruple dropout medium

supplemented with X-a-Gal (SD/-Leu/-Trp/-His/-Ade/X-a-Gal). Successful interaction was determined by the presence of blue cells on the media.

### 2.8. Construction, Expression, and Purification of the rOn-TF Plasmid

A pair of primers (TF-EcoRI-F and TF-SalI-R) with Eco R I and Sal I restriction sites were designed to amplify the sequence (55–2088 bp). The PCR products were purified and ligated into the pMD18-T vector. The recombinant pMD-18T plasmid and Pcold-ZZ were digested with Eco R I and Sal I. The expression plasmid Pcold-ZZ-TF was transformed into *E. coli* BL21 (DE3) (TransGen, China) and cultured in LB-ampicillin at 37 °C. When the culture medium $OD_{600}$ reached 0.4–0.6, isopropyl-<beta>-D-thiogalactopyranoside (IPTG) was added to a final concentration of 0.5 mmol/L and induced at 18 °C for 10 h. The bacterial solution was collected and washed three times with PBS. Lysozyme was added to a final concentration of 1 mg/mL, placed on ice for 30 min, and then centrifuged at 4 °C for 10 min. The supernatant was purified using a His-tag protein purification kit (Beyotime, Shanghai, China), desalted, and concentrated using an Amicon Ultra Centrifugal Filter (Amicon, Miami, FL, USA). The purified protein was analyzed by 10% reducing SDS-PAGE and Western blot. In addition, the recombinant His-tag (rTRX) was expressed and purified as the control for subsequent experiments.

### 2.9. Activation and Regulation of rOn-TF on NCC Activity

NCCs were isolated and purified from Nile tilapia head kidney tissues as previously described [17]. Flow cytometry analysis showed that NCCs were 80–90% positive for NCC-specific monoclonal antibody 5C6 (No: ab2778, abcam, the United States) staining (which could bind specifically to the NCCRP-1). The purified NCCs were added to 24-well microplates ($1 \times 10^6$ cells well$^{-1}$) and incubated at 25 °C.

The cells were divided into four groups with a final concentration of $1 \times 10^6$ cell/mL, namely, IgG+rTRX group, IgG+rOn-TF group, 5C6+rTRX group, and 5C6+rOn-TF group. Next, IgG (8 μg/mL), IgG (8 μg/mL), 5C6 (8 μg/mL), and 5C6 (8 μg/mL) were added to each group, respectively. Then, the cells in each group were incubated with the corresponding reagent for 1 h, and rTRX (20 μg/mL), rOn-TF (20 μg/mL), rTRX (20 μg/mL) and rOn-TF (20 μg/mL) were added to each group, respectively. Then, the cells were incubated with the corresponding reagent, and the cell samples were collected at 3 and 6 h.

RNA extraction and cDNA synthesis were performed as mentioned previously. qRT-PCR was performed to assess the molecular mechanisms of On-TF in regulating the immune responses of NCCs using various related genes. PCR was performed in a 10 μL reaction volume containing 0.5 μL of each primer (10 mM), 0.5 μL of cDNA, 5 μL of SYBR® Select Master Mix (Applied Biosystems, Waltham, MA, USA), and 3.5 μL of PCR-grade water. The PCR amplification program was as follows: 94 °C for 5 min, followed by 40 cycles at 94 °C for 10 s and 60 °C for 1 min. Melt curve analysis of amplification products was performed at 70–95 °C at the end of each PCR reaction to confirm single-product generation. The samples were run in triplicate on Applied Biosystems 7500 Real-Time PCR System (Applied Biosystems, Waltham, MA, USA). The relative expression of On-TF was calculated using the $2^{-\Delta\Delta Ct}$ method. PCR efficiency was determined, and the comparative Ct method was employed in accordance with the literature [19]. All reactions were performed with three sample replicates and three technical replicates.

### 2.10. Assay for the Killing Effect of NCCs

The fathead minnow (FHM) epithelial cells were cultured in Leibovitz's 15 medium containing 10% foetal bovine serum (Invitrogen, Waltham, MA, USA) at 25 °C. A total of $1 \times 10^6$ FHM cells were seeded in a 24-well cell plate. NCCs of Nile tilapia were cultured as described above. Briefly, the cells were divided into four groups with a final concentration of $1 \times 10^6$ cells/mL, namely, the blank group, rTRX group, 5C6+rOn-TF group, and IgG+rOn-TF group.

Next, PBS, 5C6 (8 µg/mL) and Ig G (8 µg/mL) were added to the rTRX group, 5C6+rTRX group, and IgG+rOn-TF group, respectively. Then, they were incubated with the corresponding reagent for 1 h. rTRX (20 µg/mL), rTRX (20 µg/mL), and rOn-TF (20 µg/mL) were added to each group, respectively, for 3 h. The suspending NCCs ($1 \times 10^6$ cells) in all groups were collected and co-incubated with adherent FHM cells ($1 \times 10^6$ cells) for 24 h at 25 °C. Subsequently, the plate was shaken gently, and the medium and suspending NCCs were removed. The adherent FHM cells were retained and washed with PBS thrice.

FHM cells could be observed using a microscope. The cells were resuspended in PBS and stained with trypan blue, and then dead cell percentages were calculated. Meanwhile, FHM cells were lysed using RIPA Lysis Buffer (P0013C, Beyotime, Shanghai, China), following the manufacturer's instructions. Caspase 1, 3, and 9 activities were determined using the caspase 1, 3, and 9 activity assay kit (C1101, C1115, C1157, Beyotime, Shanghai, China). All reactions were performed with three sample replicates and three technical replicates.

### 2.11. Statistical Analysis

The data are expressed as mean ± standard error (S.E.). Statistical comparison and analysis were performed using SPSS 20 software. Significant differences are indicated with * ($p < 0.05$) or ** ($p < 0.01$).

## 3. Results

### 3.1. Evaluation and Screening of Y2H Library

Y2H liver and head kidney tissue libraries were successfully constructed. Twelve clones were randomly selected from each of the two libraries to detect the size of the fragments inserted into the vector. The results showed that the inserted fragment sizes were all over 500 bp (Figure 1), indicating that the library had good construction quality and could be used for further screening. One positive clone was screened from Y2H liver and head kidney libraries using the successfully constructed bait vector PGBKT7-on-NCCRP-1, respectively (data not shown). Sequencing showed that the positive clones of the fragment were both fragments of transferrin from Nile tilapia by NCBI blast.

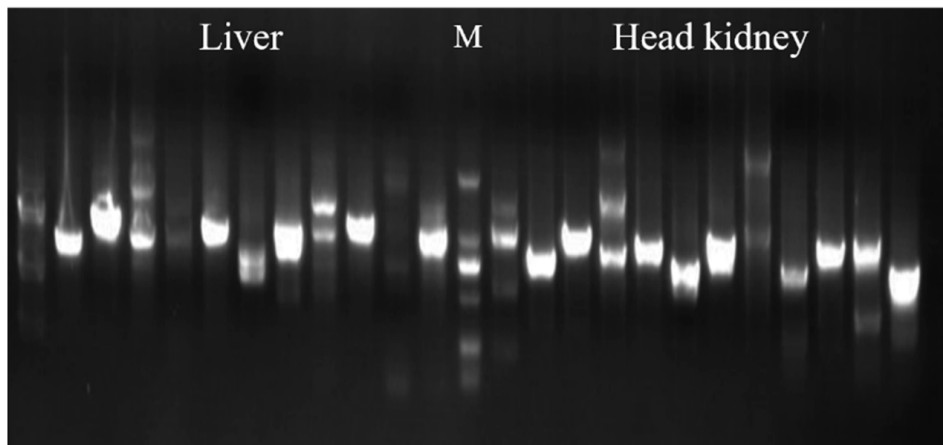

**Figure 1.** Detection of the size of the liver and head kidney libraries of Y2H. Note: M (marker): 2000, 1000, 750, 500, 250, and 100 bp.

### 3.2. Cloning and Sequence Analysis of On-TF Gene

Based on the gene fragment from the result of Y2H, the full-length cDNA sequence of On-TF was obtained (Figure 2A) via PCR technology. The complete sequence of On-TF cDNA contains a 2088 bp open reading frame encoding 695 amino acids. The protein molecular weight was determined to be 75.495 kDa. SMART program analysis showed that On-TF contained two TR_FER domains (25-343th aa and 344-686th aa) for binding iron ions (Figure 2B). Multiple sequence alignment showed that On-TF has a high sequence

homology with other species. Two TR_FER domains (25-343th aa and 344-686th aa) were conserved in the selected species (Figure 3A). The observed phylogenetic relationship implies that On-TF can be grouped with other fish TF proteins (Figure 3B). This relationship supports the established phylogeny of the selected organisms.

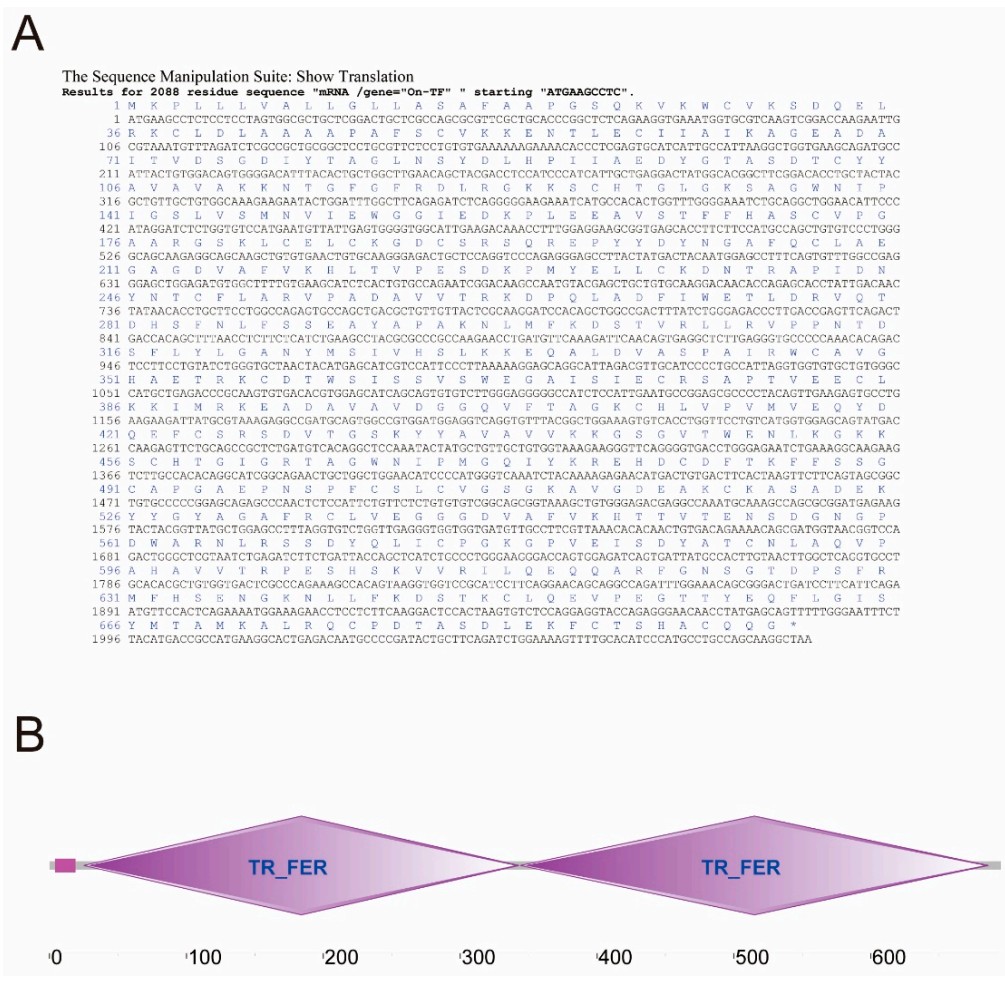

**Figure 2.** Full-length nucleotide sequences and deduced amino acid sequences of On-TF (**A**) and structure diagram of On-TF (**B**). Note: Asterisk indicates the stop codon (**A**). The TR_FER conserved domain is denoted by pink rhombus (**B**).

### 3.3. Verification of On-NCCRP-1 and On-TF Interaction

The interaction between On-NCCRP-1 and On-TF was indicated by the colors of the hybridized yeasts (Figure 4). Five yeast groups were grown in SD/-Leu/-Trp medium. For self-activation detection, pGADT7-On-NCCRP-1 and pGBKT7, and pGBKT7-On-TF and pGADT7 yeasts were grown in medium without SD/-Leu/-Trp/-His/-Ade or SD/-Leu/-Trp/-His/-Ade/X-a-Gal, suggesting that On-NCCRP-1 and On-TF themselves had no transcriptional activity and could not activate reporter genes. Moreover, a negative control did not grow in medium without SD/-Leu/-Trp/-His/-Ade or SD/-Leu/-Trp/-His/-Ade/X-a-Gal. The pGBKT7-On-TF/pGADT7-On-NCCRP-1 and pGBKT7-53/pGADT7-T1 yeasts grown on SD/-Leu/-Trp/-His/-Ade plates were white, whereas those grown on SD/-Leu/-Trp/-His/-Ade/X-a-Gal plates were blue. The results demonstrate that On-NCCRP-1 and On-TF proteins could interact directly.



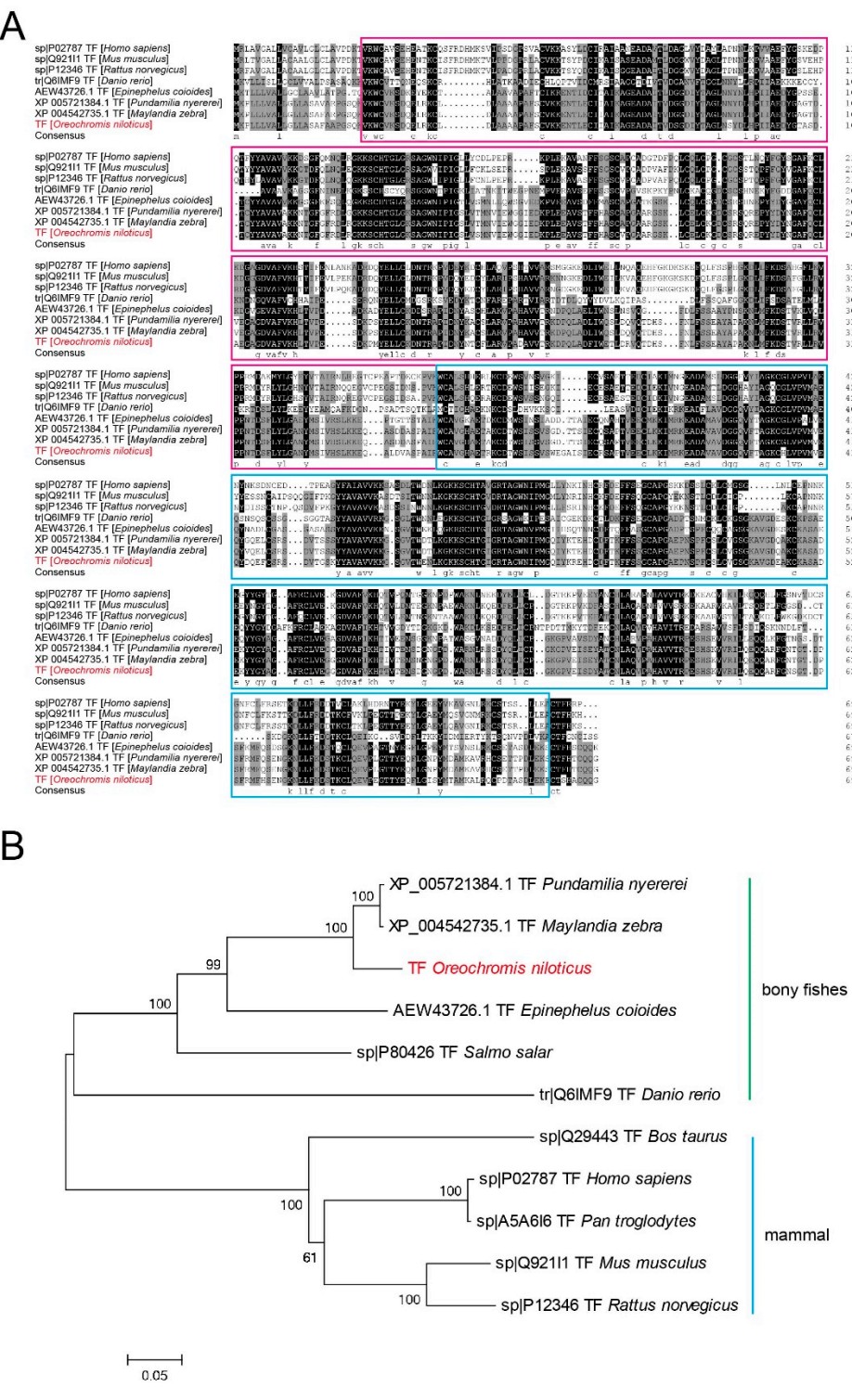

**Figure 3.** (**A**) TF multiple sequence alignment from different species. The TR_FER domains are indicated by a solid pink and a solid blue box. On-TF has a red highlight. (**B**) TF's phylogenetic tree was created using the MEGA 6 software. On-TF is highlighted in red. Note: Consensus residues are in black. Residues that are ≥75% identical amongst aligned sequences are in grey.

### 3.4. Expression, Purification, and Western Blot Analysis of rOn-TF

The expression of rOn-TF was checked by SDS-PAGE. As shown in Figure 5A, a band (~97 kDa) corresponding to On-TF-His fusion protein could be detected. The rOn-TF protein was purified; the results are shown in Figure 5B. The protein bands were relatively single. Western blot results showed that the antibodies reacted strongly with rOn-TF, and

a specific positive band was detected at 97 kDa (with anti-His-tag mouse monoclonal antibody as the primary antibody). Meanwhile, the control group with mouse IgG serum as the primary antibody had no bands. The results showed that the rOn-TF protein with His-tag was successfully expressed.

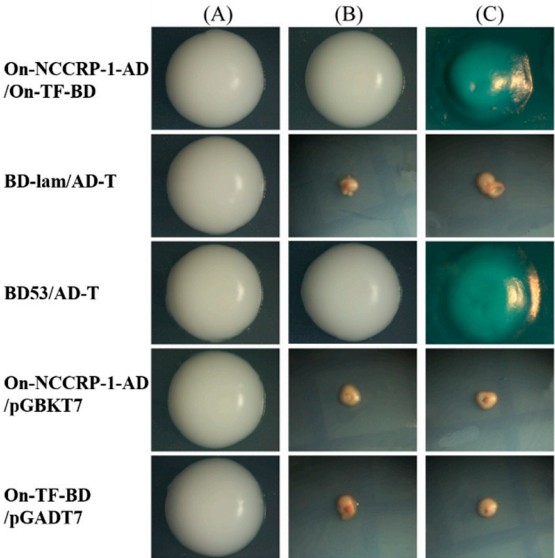

**Figure 4.** Y2H assay of the interaction between On-NCCRP-1 and On-TF. pGBKT7-On-TF/pGADT7-On-NCCRP-1 is the experimental group; pGBKT7-Lam/pGADT7-T is the negative control; pGBKT7-53/pGADT7-T is the positive control; pGADT7-On-NCCRP-1/pGADT7 and pGBKT7-On-TF/pGBKT7 are the self-activation groups. Column (**A**): Yeasts grown on plates with double dropout medium (SD/–Leu/–Trp). Column (**B**): Yeasts grown on plates with quadruple dropout medium (SD/-Leu/-Trp/-His/-Ade). Column (**C**): Yeasts grown on plates with quadruple dropout medium supplemented with X-a-Gal (SD/-Leu/-Trp/-His/-Ade/X-a-Gal).

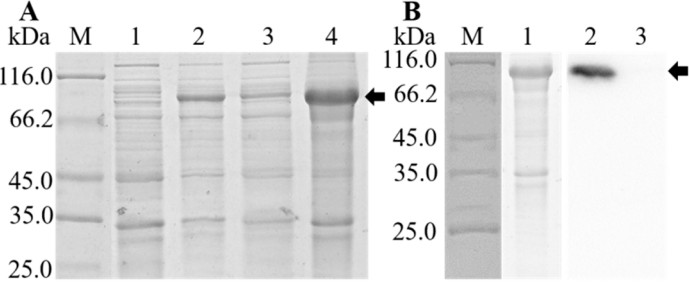

**Figure 5.** (**A**) Identification of prokaryotic protein expression of rOn-TF by SDS-PAGE. rOn-TF is shown with a black arrow. Line M: protein molecular marker; line 1, 2, 3, and 4: bacteria with Pcold-zz-On-TF without IPTG induction, bacteria with IPTG induction, bacteria soluble protein with IPTG induction, and bacteria inclusion body protein with IPTG induction, respectively. (**B**) Identification of purified rOn-TF by SDS-PAGE (line 1) and western blot (line 2 and 3). rOn-TF is shown with a black arrow; M: protein molecular marker; line 1: purified rOn-TF; line 2: experiment group (use mouse anti-his IgG); line 3: control group (use mouse IgG).

### 3.5. Effect of rOn-TF Protein on the Expression of NCC Effector Molecules

In order to determine whether rOn-TF can mediate NCC activity, rOn-TF was used to incubate the isolated NCC; and then mRNA expression level of effector molecules of NCC were detected. Effector molecules contain tumor necrosis factor (TNF) α, NCCRP-1, granzyme, FasL, cellular apoptosis susceptibility (CAS), perforin-1, Fas-associated death domain (FADD), and NK-lysin.

The results showed that some of the NCC effector molecules, such as TNF-α, granzyme FasL, CAS, and NK-lysin, were significantly upregulated in the rOn-TF group at 3 h

(Figure 6), TNF-α and CAS were significantly upregulated at 6 h, and granzyme was also slightly upregulated at 6 h, indicating that NCC was activated at 3 and 6 h after rOn-TF incubation. If the NCC was first treated with 5C6 anti-body for blocking NCCRP-1, effector molecule (except FasL at 3 h and NK-lysin at 6 h) expression mostly did not increase in the rOn-TF stimulated group at 3 and 6 h, which indicated that rOn-TF could mediate NCC activity by NCCRP-1.

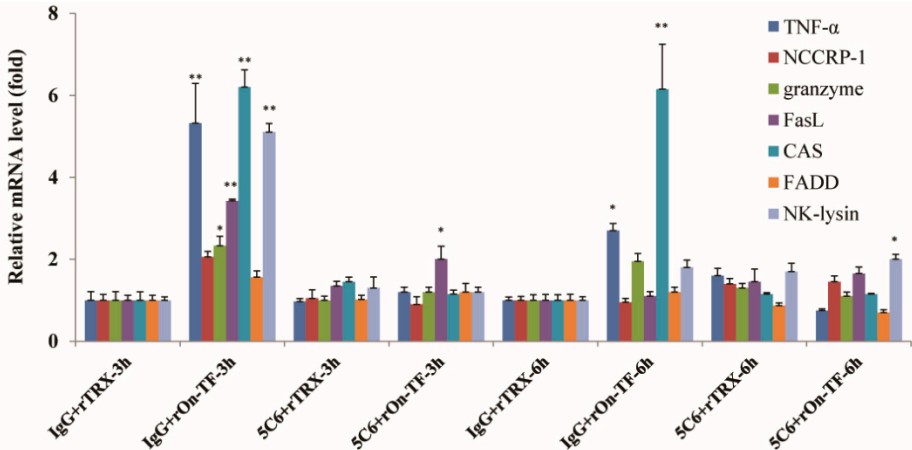

**Figure 6.** Expression of NCC-related genes after rOn-TF stimulation by qRT-PCR. The transcriptional level of NCC-related genes in the IgG+rTRX group at 3 and 6 h was set as 1. All values are the mean ± SD, $n$ = 3. Significant difference is indicated by asterisks: * $0.05 > p > 0.01$, ** $p < 0.01$.

*3.6. Effect of rOn-TF Protein on the Cytotoxicity of NCC*

To determine whether rOn-TF protein enhances the toxic effect of NCCs, the NCCs were incubated with rOn-TF and then added a certain proportion of FHM cells. After that, the cell-killing activity of NCC to FHM cells was upregulated. The live cell count in different groups showed a significant discrepancy. Then, we calculated the dead cell percentage of each group, and the results were as follows: 11.8% in the blank NCC group, 21.3% in the rTRX NCC group, 26.5% in the 5C6+rOn-TF NCC group, and 75.5% in the IgG+rOn-TF NCC group (Figure 7).

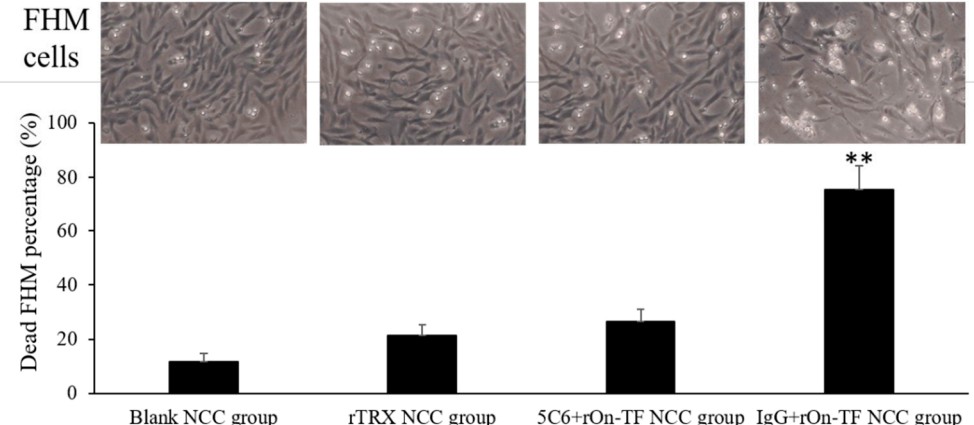

**Figure 7.** Cell-killing activity of NCCs on FHM cells. Photographs of FHM cells (upper) and corresponding dead cell percentage of FHM cells (lower). Data are shown as mean ± SE, and the significant difference is indicated by asterisks: ** $p < 0.01$.

Meanwhile, the killing effect of NCCs on FHM cells was determined by detecting the activities of caspase-1, 3, and 9. The results are shown in Figure 8. No significant difference in the activities of caspase-1, 3, and 9 were observed between the blank NCC+FHM group and rTRX NCC+FHM group, indicating that the labelled protein did not increase the

toxic effect of NCC. The activities of caspase-1 and 3 in the rOn-TF treatment group were significantly higher than those in the blank NCC+FHM and rTRX NCC+FHM groups, indicating that rOn-TF can significantly improve the toxic effects of NCC. In addition, if 5C6 antibody was added to block the receptor binding site of On-NCCRP-1, caspase-1 and 3 were significantly inhibited. This suggests that rOn-TF can activate the killing effect of NCC through On-NCCRP-1, and the 5C6 antibody can effectively block On-NCCRP-1. However, the activity of caspase-9 did not increase significantly; there was only a slight increase in the experimental group.

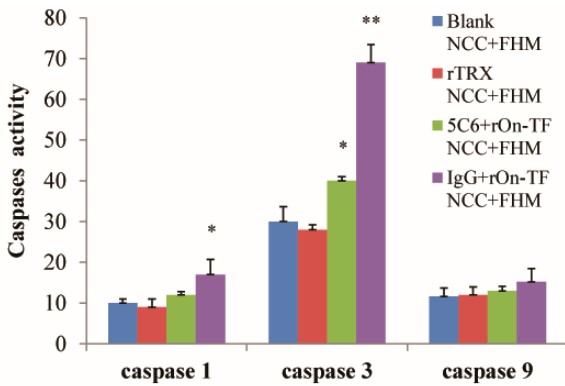

**Figure 8.** Effect of caspase-1, 3, and 9 activities in FHM cells by rOn-TF-activated NCC. Value represented as mean ± S.E, $n = 3$. Significant difference is expressed as follows: * $0.05 > p > 0.01$, ** $p < 0.01$.

## 4. Discussion

NCC is a kind of fish specific lymphocyte which plays an essential role in the innate immune response. NCCRP-1, as an important receptor protein molecule on the surface of NCCs, participates in the regulation of NCC activity. However, it is unclear which interacting molecules could bind to NCCRP-1 and then regulate the activity of NCC. Yeast two hybrid (Y2H) technology is widely used in the study of protein interaction [20,21]. In this study, we first constructed a Y2H library of tilapia liver and head kidney, and then used NCCRP-1 as a bait vector to screen two Y2H libraries so as to obtain possible interaction proteins. The quality of the library is very important for the success of the experiment. In this study, the length of the fragments inserted into the vector of the two libraries constructed was detected by randomly selected positive clones. It was found that the lengths of the fragments were all more than 500 bp, indicating that the quality of the library meets the subsequent experiments. Then, we obtained the interaction protein called transferrin (TF) from both libraries, which indicates that TF is likely to be the interaction molecule of NCCRP-1 in tilapia.

Since the TF obtained by screening is a fragment, we next cloned and obtained the ORF sequence of TF by PCR based on the comparison of NCBI and transcriptome data, and analyzed the sequence. On-TF protein showed that there were 695 amino acids. It was found that this molecule contained two TR_FER domains (25-343th aa and 344-686th aa), which may be responsible for the iron-transporting function [22–24]. Multiple sequence alignment showed that On-TF has a high sequence homology with other species. Two TR_FER domains (25-343th aa and 344-686th aa) were conserved in the selected species (Figure 3A). The results were the same as those of Chinese black sleeper (*Bostrichthys sinensis*) and common carp (*Cyprinus carpio* L.) [25,26]. This result indicates that TF in different species have similar functions. Phylogenetic analysis found that all of the bony fish gathered in a cluster, and mammals gathered in another branch. This relationship supports the established phylogeny of the selected organisms.

Although Y2H technology is widely used in the study of protein interaction, it can produce many false positives [21]. After obtaining the full length of On-TF and understanding the characteristics of this gene, we further verified interaction of On-TF and

On-NCCRP-1. Here, we still used the Y2H technology [27], but the vector linking On-TF and On-NCCRP-1 was interchanged, and a series of control groups were set up to reduce the occurrence of false positives. The results are also consistent with those of the previous stage. Blue colonies were found on the screening medium, indicating that On-TF could interact with On-NCCRP-1. These results further suggest that On-TF molecules may bind with On-NCCRP-1 in vivo to regulate the immune activity of NCC.

To test the above hypothesis, we next obtained the recombinant fusion protein rOn-TF. The isolated NCCs were incubated with rOn-TF to detect the mRNA expression level of NCC-related effector genes. The results showed that some effector genes (especially at 3 h), such as TNF-$\alpha$, granzyme, FasL, CAS, perforin-1 and NK-lysin, were significantly upregulated in the rOn-TF stimulation group after incubation. Tilapia TNF-$\alpha$ can protect NCCs from apoptosis induced by activation and promote the expression of granzyme in NCCs [28]. At the same time, tilapia TNF-$\alpha$ can cause cytotoxic effects. TNF-$\alpha$ can be constitutively expressed by tilapia NCCs after activation. TNF-$\alpha$ binds to the membrane surface receptor TNFR-1 to exert its cytotoxic effect, and has a strong scavenging function on cancer and virus-infected cells [28,29]. The pro-apoptotic protease granzyme enters target cells, resulting in the death of cancer cells, pathogen-infected cells, and bacteria or restriction of virus replication [30,31]. CAS is involved in multiple cellular mechanisms associated with cell proliferation and cell death [32]. In addition, activated-NCCs in tilapia will secrete lots of effector molecules such as perforin-1 and NK-lysin to kill target cells [33]. Thus, NCC may respond to extracellular signal stimulation, which is similar to our previous results. rOn-NKEF protein effectively activates NCC and significantly upregulates the expression of its related effectors [33]. In addition, if On-NCCRP-1 was blocked by 5C6 antibody in advance, the related genes would not be upregulated at 3 and 6 h. Therefore, we concluded that rOn-TF mediated signal transmission through On-NCCRP-1 and activated the activity of NCC. Next, rOn-TF-activated NCCs were incubated with FHM cells. A large number of deaths were observed in FHM cells, which further indicated that the rOn-TF protein can significantly enhance the killing ability of NCC to FHM cells.

In the process of apoptosis, the activity of the protease family cystaine—requiring aspartate protein (caspase)—is significantly increased in vivo [34]. Therefore, cell apoptosis can be judged by detecting the activity of caspases. In this study, the activities of caspase-1 and 3 in cells treated with rOn-TF were significantly higher than those in the control group, indicating that rOn-TF could significantly improve the killing activity of NCC to FHM cells, leading to the activation of FHM apoptotic signal. When On-NCCRP-1 was blocked, the activities of caspase-1 and 3 were effectively suppressed, which further proves that On-TF enhanced the killing activity of NCC by binding to the receptor NCCRP-1 on NCC.

## 5. Conclusions

This study obtained an interaction protein of On-NCCRP-1 (Nile tilapia transferrin protein [On-TF]) and analyzed its sequence characteristics. On-TF could upregulate the expression of killer effector molecule of NCC by On-NCCRP-1, and On-TF-activated NCCs showed a significantly improved ability to kill FHM cells. The results provide a more theoretical basis for understanding the regulation mechanism of NCC activity.

**Supplementary Materials:** The following supporting information can be downloaded at https://www.mdpi.com/article/10.3390/fishes7050253/s1. Table S1: The primers used in this research. Table S2: GenBank accession numbers of relative genes of tilapia used in this study.

**Author Contributions:** Conceptualization, Y.H.; methodology, Y.H. and Z.Z.; formal analysis, Y.H. and Z.C.; investigation, Z.C., R.X. and P.W.; resources, Z.C., R.X., P.W. and B.W.; writing—original draft preparation, Y.H.; writing—review and editing, J.C. and B.W.; supervision, J.J.; funding acquisition, Y.H. and J.J. All authors have read and agreed to the published version of the manuscript.

**Funding:** This research was funded by the National Natural Science Foundation of China (No. 32002426, 32073006) and the Natural Science Foundation of Guangdong Province (No. 2022A1515010553).

**Institutional Review Board Statement:** The study was conducted in accordance with the Declaration of Helsinki and approved by the Ethics Committee of the Guangdong Ocean University (Date: 10 May 2019).

**Informed Consent Statement:** Not applicable.

**Data Availability Statement:** Data are contained within the article or figurementary Materials.

**Acknowledgments:** We extend our thanks to Zhe Zheng and Kwaku Amoah for reviewing this manuscript.

**Conflicts of Interest:** The authors declare no conflict of interest.

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
