# Peer review of "Transferrin Mediated NCC Killing Activity through NCCRP-1 in Nile Tilapia (Oreochromis niloticus)"

_fishes, doi:10.3390/fishes7050253_

Round 1

Reviewer 1 Report

This paper shows that binding of non-specific cytotoxic cell receptor protein 1 (NCCRP-1) to transferrin (TF) is required for the activation of fish non-specific cytotoxic cells (NCCs). Fish NCCs are thought to have a role similar to that of mammalian NK cells, but the details have not been elucidated, and this paper provides very useful information for future studies of NCCs.

On the other hand, there are many typographical errors and ambiguous explanations in the manuscript, and I have decided that the manuscript cannot be published in Fishes as it is. Many of the primers used in the experiments also did not include basic information that should have been included, such as accession numbers and citation sources. In addition, although the experiments using mAb 5C6 to inhibit the function of NCCRP-1 are very interesting, there are not sufficient descriptions and experiments on how mAb 5C6 is produced and what antigens it recognizes. The authors need to confirm the interaction between NCCRP-1 and mAb 5C6 by methods other than functional inhibition of the receptor, for example, by confirming the reactivity of mAb 5C6 with cells transfected with the NCCRP-1 gene.

Reviewer 2 Report

The author identified transferrin of the tilapia to be a ligand of NCCRP. I have some questions.

Although the author prepared recombinant TF, why they did not confirmed the binding activity between NCCRR-1 and TF in vitro (cf pull down assay)? 

Please describe the details on monoclonal antibody 5C6. This antibody has not been described in ref 18. Additionally, the author should check whether the antibody bind to rTF non-specifically.

Why the author choose the concentration of rTF in this explain? The concentration of 20 microg/mL seems to be high. The author should perform the concentration dependent experiments.

Round 2

Reviewer 1 Report

I have no further comments to make on this paper.

Reviewer 2 Report

No comments.